# Identifying gaps in global evidence for nurse staffing and patient care outcomes research in low/middle-income countries: an umbrella review

Abdulazeez Imam ,[1,2] Sopuruchukwu Obiesie,[3] Jalemba Aluvaala ,[1,4] Jackson Michuki Maina ,[1] David Gathara,[5,6] Mike English [1,2]

For numbered affiliations see end of article.

**Correspondence to**
Dr Abdulazeez Imam;
abdulimam2001@yahoo.com

## ABSTRACT

**Objective** To identify nurse staffing and patient care outcome literature in published systematic reviews and map out the evidence gaps for low/middle-income countries (LMICs).

**Methods** We included quantitative systematic reviews on nurse staffing levels and patient care outcomes in regular ward settings published in English. We excluded qualitative reviews or reviews on nursing skill mix. We searched the Cochrane Register of Systematic Reviews, the Joanna Briggs Institute Database of Systematic Reviews and Implementation Reports, Medline, Embase and Cumulative Index to Nursing and Allied Health Literature from inception until July 2021. We used the A Measurement Tool to Assess Systematic Reviews -2 (AMSTAR-2) criteria for risk of bias assessment and conducted a narrative synthesis.

**Results** From 843 papers, we included 14 in our final synthesis. There were overlaps in primary studies summarised across reviews, but overall, the reviews summarised 136 unique primary articles. Only 4 out of 14 reviews had data on LMIC publications and only 9 (6.6%) of 136 unique primary articles were conducted in LMICs. Only 8 of 23 patient care outcomes were reported from LMICs. Less research was conducted in contexts with staffing levels that are typical of many LMIC contexts.

**Discussion** Our umbrella review identified very limited data for nurse staffing and patient care outcomes in LMICs. We also identified data from high-income countries might not be good proxies for LMICs as staffing levels where this research was conducted had comparatively better staffing levels than the few LMIC studies. This highlights a critical need for the conduct of nurse staffing research in LMIC contexts.

**Limitations** We included data on systematic reviews that scored low on our risk of bias assessment because we sought to provide a broad description of the research area. We only considered systematic reviews published in English and did not include any qualitative reviews in our synthesis.

**PROSPERO registration number** CRD42021286908.

## BACKGROUND

Donabedian in his seminal work on quality of care describes a structure–process–outcome framework.[1] In his framework,

---

## STRENGTHS AND LIMITATIONS OF THIS STUDY

⇒ This umbrella review comprehensively searched across five electronic databases for systematic reviews on the association between nurse staffing and patient care outcomes.

⇒ We conducted our umbrella review using guidance from the Joanna Briggs Institute, which is the standard for conducting umbrella reviews.

⇒ We had a broad focus to describe the global evidence and identify gaps for low/middle-income countries and so we included reviews that scored low on our risk of bias scores.

⇒ We included only systematic reviews published in English and did not include any qualitative reviews.

---

structures present within health services influence the processes of care which in turn are likely to affect care outcomes.[1] A key element of health service structure is staffing, and nurses who represent a large percentage of hospital staffing are thus likely to have significant effects on hospital patient outcomes.[2] Nurses play multiple, crucial roles such as planning, delivering and coordinating care, and represent a key part of the hospitals' surveillance system in detecting adverse patient events.[3]

An adequate nursing workforce is central to the delivery of quality patient care. Research that comes from mainly high-income countries (HICs) demonstrates that poorer ward nurse staffing is associated with negative patient care outcomes, for example, increased risk of patient mortality, prolonged hospital stay and an increased risk of hospital-acquired complications.[4–14] In these studies, staffing levels have been measured using metrics which measure the number of nurses or nursing care hours delivered by them relative to patient numbers or proxy measures for patient numbers, for example, hospital

beds.[4–14] This has led to a shift towards improved staff-to-patient ratios or minimum nurse staffing standards to guide ward nurse staffing levels in many HICs. In 1999, California state adopted a bill mandating the state Department of Health Services to enforce minimum staffing ratios across hospitals within the state, and this has been the focus of multiple research studies investigating the effects on the quality of patient care.[15–18] Shortly after this, in 2002, the state of Western Australia used the nursing hour per patient day (NHPPD) method which classifies wards into seven categories based on patient care acuity and complexity to determine relative nurse staffing needs.[19 20] In other countries such as the UK, the Royal College of Nursing also provides minimum staffing guidance for wards, for example, paediatric wards are required to maintain a minimum ratio of one registered nurse to four children with better ratios applied to high dependency and intensive care settings.[21]

The evolution and the evidence for nurse staffing in low/middle-income countries (LMICs) are less clear. In more resource-challenged LMIC settings, nurse staffing ratios might be as low as 1 nurse caring for over 25 patients on a shift.[22 23] For such settings, the WHO has promoted the Workload Indicator Staffing Needs (WISN) planning tool which relies on data from health information systems.[24] The central focus of the WISN tool is the workload of the average health worker and the time to carry out their activities as defined by experts.[24] It aims to provide context-specific estimates of workforce requirements that are based on service types and complexity within local health facilities. However, the WISN approach often suggests workforce expansion that seems unachievable for countries due to the major increases in rates of production and financing required, highlighting the practical realities of human resource costs which can represent up to two-thirds of health service budgets.[25 26] It is highly likely that the current extreme staffing ratios in LMICs contribute to poor patient outcomes and overall poor-quality care, and research to understand this is crucial.

A recent systematic review on the impact of nurse staffing on patient and nurse workforce outcomes in LMICs demonstrated limited and poor-quality evidence on the role of nurse staffing and patient care outcome in LMICs.[27] A crucial next step to guide the conduct of research in these settings is to examine the existing evidence gaps for research in LMICs. One way to do this would be to compare the evidence in these settings with HICs which have comparatively greater volume of research. An immediate challenge to this is the increasing and expanding volume of literature in this area of research which has likely resulted in more focused systematic reviews over the last decade.[10 27 28] More recent reviews have either focused on specific patient populations or regions of the world,[27] or specific patient care outcomes,[10 28] contrasting earlier reviews which were more broad based.[14 29]

Umbrella reviews synthesise the information from systematic reviews and can serve as a more efficient synthesis method for areas where large volumes of research have been conducted, integrating research to provide broader knowledge on specific topics.[30] Using this method, we appraised the existing global evidence related to nurse staffing and patient care outcome research in published systematic reviews and compared the evidence available for LMICs with HICs, so we could identify the existing evidence gaps for research in LMICs.

## Aim and objectives

The primary objective of this umbrella review is to identify the evidence gaps for LMICs in the existing literature that investigates the association between hospital nurse staffing and patient care outcomes. We will address the following research questions:

1. Where are studies within published reviews conducted and what proportion of these are carried out in LMICs?
2. What patient care outcomes are reported across reviews and how do reported outcomes differ between HIC and LMIC studies?
3. What is the range of nurse staffing levels that have been researched across acute care settings and how do these differ between LMICs and HICs?

## METHODS

### Research design

We conducted an umbrella review using guidance from the Joanna Briggs Institute (JBI) for the preparation and conduct of the review.[30] Our review protocol was registered with the International Prospective Register of Systematic Reviews (registration number: CRD42021286908) and has been published.[31]

### Data sources and search strategy

To identify published systematic reviews that reported on the association between nurse staffing and patient care outcomes, we conducted a systematic search of five electronic databases from their inception up until July 2021. These were the Cochrane Register of Systematic Reviews, the JBI Database of Systematic Reviews and Implementation Reports, Medline, Embase and Cumulative Index to Nursing and Allied Health Literature. Our search strategy for this is detailed in online supplemental file 1 and has previously been published.[31]

We included some of the following keywords and their synonyms: nurse, nursing, outcome, quality, missed nursing, mortality and identified some key Medical Subject Headings (MESH) terms (online supplemental file 1). We combined these using Boolean operators 'AND' and 'OR' where appropriate. We also searched the reference list of our included systematic reviews to identify other reviews.

### Selection of systematic reviews

We included quantitative systematic reviews that: (a) summarised literature on nurse staffing levels and patient care outcomes in regular ward settings; (b) were

conducted in English language (due to limitations in translation among the research team).

Reviews were excluded if they: (a) were conducted in a non-hospital setting; (b) only summarised studies conducted in non-regular ward settings, for example, intensive care units; (c) reported exclusively on non-patient care outcomes, such as nurse outcomes (for example, burnout and nurse satisfaction); (d) reported on other nurse staffing characteristics other than staffing levels, for example, skill mix (a measure of nurse organisation); (e) were qualitative and mixed-method reviews.

## Population

The participants in the included systematic reviews were patients admitted to regular hospital ward settings. For reviews reporting on patients in both standard ward settings and intensive care units, we include these but only summarised and reported on specific data which focused on our inclusion criteria.

## Exposure

Our exposure for this review is the level of nurse staffing and thus reflects nursing metrics that focus on nursing numbers or nursing time available to patients, for example, nurse-to-patient ratios or NHPPD and not on how the nursing workforce is organised.[31]

## Outcome

Our outcome of interest was patient care outcomes, for example, mortality and healthcare-associated infection. For reviews that reported mixed outcomes, for example, reviews summarising literature on both nurse and patient outcomes, we reported only on patient care outcomes.

## Screening

References were managed using the Zotero reference software.[32] We performed initial deduplication in Zotero and a second round of deduplication in Microsoft Excel. Following which we exported our final set of articles to Rayyan, a web-based application for screening,[33] where two reviewers, AI and SO, independently screened the titles and abstracts of all identified systematic reviews for eligibility. They then read full texts of selected articles and agreed on a final set of papers.

## Quality assessment

We evaluated each systematic review for risk of bias using the A Measurement Tool to Assess Systematic Reviews-2 (AMSTAR-2) criteria.[34] This tool comprises 16 appraisal questions rated 'yes' when a criterion is fulfilled; 'no' when it is not fulfilled and 'partial yes' when it is somewhat fulfilled (this is based on explicitly stated criteria). The AMSTAR criteria cover important questions such as whether the Population, Intervention, Comparator and Outcome (PICO) elements were clearly described by the review, whether the conduct of study selection, data abstraction and risk of bias were conducted by at least two reviewers and the use of a comprehensive search strategy.[34] The quality appraisal was performed independently by AI and SO, and disagreements were managed through discussions.

## Data extraction

Both reviewers (AI and SO) independently extracted data from the final set of systematic reviews using a predesigned data abstraction tool. These data included the review publication year, objectives, reported patient care outcomes, number and origin of review studies that reported on nurse staffing and patient outcomes in regular ward care settings.

Summary data on the nurse staffing metrics were also retrieved from the individual paper summaries within systematic reviews. Where this information was unavailable as a summary within the reviews, we abstracted them directly from the individually referenced articles. AI abstracted these data and SO cross-checked a random 40% of this.

## Data synthesis

Our umbrella review findings are presented in narrative form using tables and figures. We tallied the countries of origin of the individually referenced papers within each review and classified these into LMIC and non-LMIC using the World Bank country and lending group classification system as of 23 December 2021.[35] We also summarised and tallied all reported patient care outcomes from the included systematic reviews and made comparisons made between LMICs and HICs.

We identified and collated summary statistics (range, mean or median) for nurse staffing metrics reported in individually referenced papers using tables and bar charts where these were reported. Because of the heterogeneity of retrieved metrics, a meta-analysis was not possible.

## Patient and public involvement

There was no patient or public involvement in the design and conduct of this study.

## RESULTS

### Search results

Our initial search of databases identified 1365 articles. We identified and excluded 522 duplicate articles and screened the title and abstract of 843 remaining articles. Of the 33 systematic reviews that met our criteria for full-text reviews, we included 14.[10 14 19 27–29 36–43] The Preferred Reporting Items for Systematic Reviews and Meta-Analyses flow chart in figure 1 provides a summary of our screening process, while online supplemental file 2 shows the list of excluded systematic reviews and the reasons for their exclusion.

### Description of included reviews

We included 14 systematic reviews in our final analysis. The reviews covered periods from as early as 1980 up until 2020 (table 1). The reviews were conducted on studies from predominantly medical, surgical or adult wards with

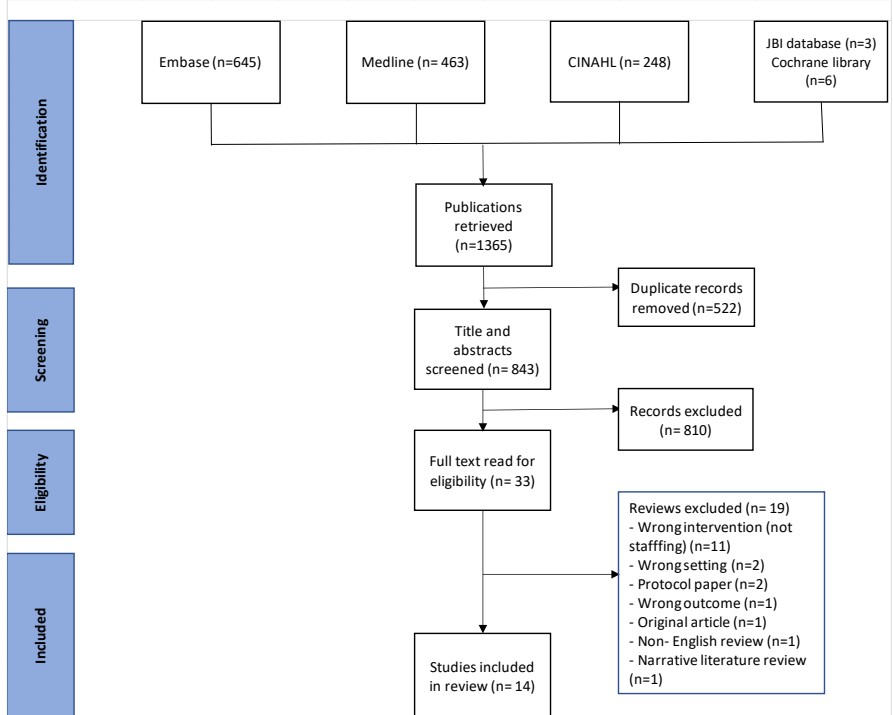

**Figure 1** – PRISMA diagram for search

**Figure 1** Preferred Reporting Items for Systematic Reviews and Meta-Analyses diagram for search.

only one review conducted on those from paediatric ward settings.[39] The reviews included were between 5 and 54 primary studies. Of the individual studies within reviews, between 12.5% and 100.0% met our inclusion criteria (table 1). This was because some reviews in addition to summarising studies that investigated the effect of nurse staffing on patient care outcomes also included those that reported other outcomes, for example, nurse outcomes.[27] Other reviews also included studies conducted in non-regular ward settings such as intensive care unit settings,[28 38] or included a mixture of studies investigating nurse staffing and other interventions such as nurse skill mix.[14] In online supplemental file 3, for each review, we have provided a list of primary studies we did not report on and the reasons for this. There was some overlap in the primary articles summarised by the reviews, some papers were referenced in as many as six reviews (online supplemental file 4). In total, 136 unique primary studies were identified across all systematic reviews (online supplemental file 4).

### Quality assessment of included reviews
We used the AMSTAR-2 quality assessment tool to appraise individual systematic reviews.[34] As shown in figure 2, question 2 (Q2), 11 out of 14 of the reviews did not refer to a study protocol or provide explicit statements of their review methods being established before the review conduct. Most reviews (12 out of 14) did not provide a list of excluded studies with justifications on reasons for excluding these studies (figure 2, Q7). Discussions

on how the risk of bias assessments of individual studies might have impacted the review results were also limited (figure 2, Q13).

### Origin of primary studies and proportion of studies in systematic reviews conducted in LMIC settings
Of the 136 unique studies, 94 (69.1%) were carried out in the USA. Only five LMIC countries, Thailand, Brazil, Lebanon, China and Ethiopia, accounted for 9 (6.6%) of 136 primary studies (online supplemental file 5). Studies from HICs came from more diverse locations. These studies were conducted in 16 other settings in addition to the USA (online supplemental file 5). Four out of 14 of our included systematic reviews reported data on our subject of interest (nurse staffing and patient care outcomes) from an LMIC (table 1). One review restricted to LMIC settings identified only six articles, while three reviews were primarily restricted to HICs (table 1).

### Reported patient care outcomes and differences across research settings (LMICs and HICs)
In total, the included reviews reported 23 patient care outcomes. Each review reported between 1 and 15 outcomes, and the most frequently reported (eight times each) were patient mortality, pressure ulcers and the incidence of hospital-acquired infections (figure 3).

Only 8 of the 23 patient care outcomes were reported in an LMIC setting. These were missed nursing care, mortality, pressure ulcers, length of stay, treatment errors, hospital-acquired infections, falls and hospital-acquired

**Table 1** An overview of the included systematic reviews showing the review objective and geographical locations where the reviews of primary studies were conducted

| First author (year) | Period of review | Research setting | Number of primary studies included | Number of primary studies describing nurse staffing levels and patient outcomes in non-ICU settings* | Geographical locations of studies and frequency | Number of low/middle-income countries included (%) | Number of high-income countries included (%) |
|---|---|---|---|---|---|---|---|
| Griffiths (2018)[10] | 2006–2016 | Adult hospital inpatient wards | 18 | 17 | Kuwait: 1, Europe: 3, UK: 1, Sweden: 1, USA†: 7, Lebanon†: 1, Switzerland: 1, Italy: 1, South Korea: 2 | 1 (5.6) | 17 (95.4) |
| Assaye (2020)[22] | UR– 2019 | Acute care hospital settings | 27 | 6 | Brazil: 1, Thailand: 2, China: 1, Ethiopia: 1, Lebanon: 1 | 6 (100.0) | 0 (0.0) |
| Thungjaroenkul (2007)[40] | 1990–2006 | ICUs, medical and surgical units | 17 | 5 | USA: 5 | 0 (0.0) | 5 (100.0) |
| Twigg (2021)[19] ‡ | 2000–2020 | General medical, surgical, step-down units, emergency departments, intensive care and nursing homes | 22 | 7 | USA: 6, Australia: 1 | 0 (0.0) | 7 (100.0) |
| Shin (2019)[43] | 2000–2018 | Medical and surgical units | 19 | 17 | USA: 10, South Korea: 2, Finland†: 1, Netherlands†: 1, Lebanon: 1, Belgium†: 1, China: 1, Japan: 1 | 2 (11.1) | 16 (88.9) |
| Lang (2004)‡ | 1980–2003 | Acute care hospitals | 43 | 24 | USA: 24 | 0 (0.0) | 24 (100.0) |
| Lankshear (2005)[42] | 1990–2004 | Acute care hospitals | 22 | 19 | USA: 18, Canada: 1 | 0 (0.0) | 19 (100.0) |
| Kane (2007)[14] | 1990–2006 | Acute care hospitals | 28 | 17 | USA: 17 | 0 (0.0) | 17 (100.0) |
| Bourgon (2019)[38] | 1996–2018 | Surgical units | 44 | 28 | USA: 17, Europe: 2, South Korea: 1, UK: 2, New Zealand: 1, Australia: 2, Belgium: 2, Japan: 1 | 0 (0.0) | 28 (100.0) |
| Wilson (2011)[39] | 1993–2010 | Paediatric wards | 8 | 5 | USA: 2, France: 1, Canada: 1, UK: 1 | 0 (0.0) | 5 (100.0) |
| Engineer (2016)[41] ‡ | 2000–2012 | Acute care hospitals | 16 | 2 | USA: 2 | 0 (0.0) | 2 (100.0) |
| Stalpers (2015)[36] | 2004–2012 | Acute care hospitals | 29 | 16 | USA†: 12, Belgium†: 1, New Zealand: 1, Australia: 1, UK: 1, Sweden: 1 | 0 (0.0) | 17 (100.0) |
| Mitchell (2018)[28] | 2000–2015 | Hospital wards | 54 | 19 | USA: 13, Thailand: 1, Australia: 1, Taiwan: 1, Canada:2, multicountry: 1 | 1 (4.3) | 18 (95.7) |

Continued

**Table 1** Continued

| First author (year) | Period of review | Research setting | Number of primary studies included | Number of primary studies describing nurse staffing levels and patient outcomes in non-ICU settings* | Geographical locations of studies and frequency | Number of low/middle-income countries included (%) | Number of high-income countries included (%) |
|---|---|---|---|---|---|---|---|
| Hill (2017)[37] | 1994–2017 | Acute care hospitals | 5 | 5 | USA: 3, UK: 2 | 0 (0.0) | 5 (100.0) |

*These are the number of studies that met the original review criteria, that is, conducted in a regular ward setting on the role of nurse staffing and patient care outcomes.
†One study conducted across both countries.
‡Restricted search criteria to high-income countries.
ICU, intensive care unit; UR, unreported.

injuries (figure 3). Other outcomes such as postoperative complications, cardiac arrests, deep venous thrombosis, failure to rescue, unplanned extubations and incidence of restraint use were not reported in LMICs (figure 3).

## Measures of nurse staffing and reported staffing range across LMICs and HICs

There were 29 different reported nurse staffing metrics across the individual studies within our included systematic reviews. The most frequently reported metrics were the NHPPD and the patient-to-nurse ratio per shift which were reported by 48 and 34 primary studies, respectively (online supplemental file 6). There was marked heterogeneity in how both metrics were reported by papers. Some reported absolute values, mean, median, ranges, percentiles or categories. For those reporting either a range, mean or median, we have summarised their values in figures 4 and 5.

Figure 4 depicts patients per nurse for nine studies (eight HICs and one LMIC study). Studies from HICs reported best staffing ratios (smallest patient per nurse ratio) ranging between four and seven patients per nurse,[7 44–50] and this was in contrast to the sole LMIC study from Brazil which reported a best ratio of nine patients per nurse.[51] Similarly, the worst ratios documented in studies ranged between 9 and 18 patients per nurse within HICs in contrast to 27 patients per nurse in the LMIC study (figure 4).[51]

All studies which reported the mean patient-to-nurse ratio per shift were in HICs, and 16 of 18 reported mean values below 10 (figure 5).[44 45 49 50 52–62] For studies reporting total mean NHPPD, the three reported LMIC studies (from Thailand and Lebanon) had the lowest values,[63 64] and these were almost 2.5-fold below studies conducted in some HICs (figure 5).[65–70]

## DISCUSSION

This umbrella review of published systematic reviews on nurse staffing and patient care outcomes examined the evidence gaps for LMICs by appraising the global evidence. We found a dearth of evidence from LMICs with the link between nurse staffing and quality exploring a narrower range of patient care outcomes.

Of the 14 included systematic reviews in this study, 10 (71.4%) had no data from an LMIC setting. Across all 14 reviews, only 9 of the 136 unique primary studies included were from LMICs and these data came from five countries (Thailand, Lebanon, China, Brazil and Ethiopia),[51 63 64 71–76] with one from Africa (Ethiopia).[76] A recent LMIC-focused systematic review found only six nurse staffing and patient care outcome papers that were graded as providing low-quality evidence.[27] With 90% of the global deficit of nurses concentrated in LMIC settings,[2] research is needed in such settings to examine the impact these shortages have on the quality of hospital care. Ideally, these studies would examine the impact of improving staffing numbers on the overall quality

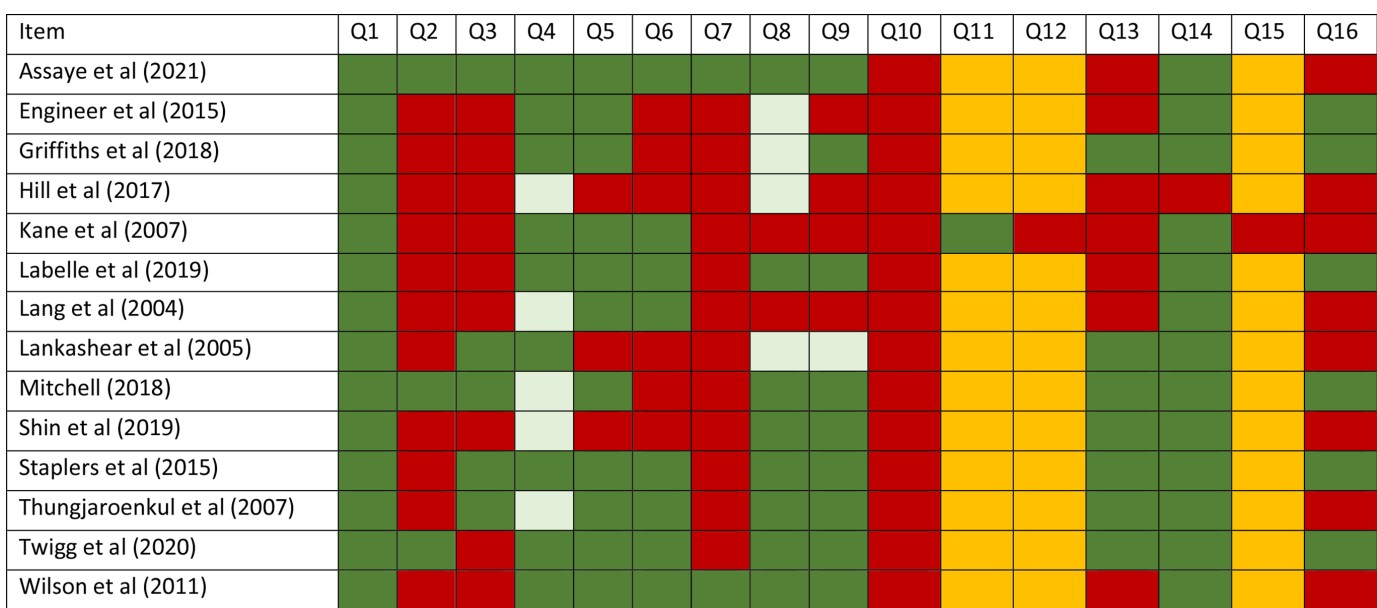

**Figure 2** Risk of bias assessment using the AMSTAR-2 checklist (key: red–no, green–yes, lighter green–partially yes, yellow–not applicable).

of patient care that is provided and patients' health outcomes. These data are critical to highlight the effects of the current staffing shortages and to guide policy-makers and managers in developing appropriate staffing policies.

The data that currently exist illustrate the negative effects of low nurse staffing on quality of care but are almost exclusively from HICs.[7 53 77] Traditionally, these areas already have better staffing ratios than LMICs and yet relatively small absolute or relative improvements in

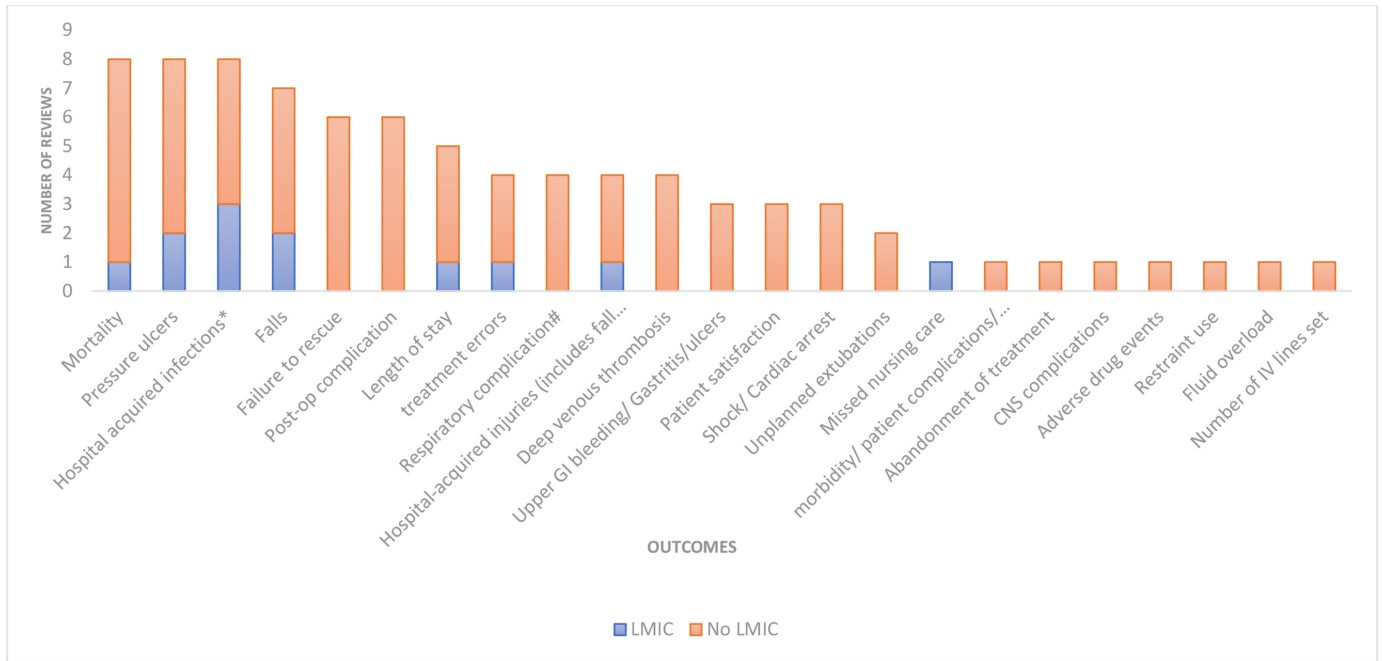

**Figure 3** Stacked bar chart showing the range of patient care outcomes reported across reviews and the number of reviews which report this in an LMIC study (blue: number of reviews reporting the outcome from an LMIC study; orange: number of reviews reporting the outcome from only non-LMIC studies). *Hospital-acquired infections include pneumonia, UTI, central line-associated blood stream infections, nosocomial infections, wound infections. #Respiratory complications include pulmonary compromise/respiratory failure/pulmonary embolism. The least reported patient care outcomes were missed nursing care, patient adverse events, abandonment of treatment, CNS complications, adverse drug events, restraint use, fluid overload and the number of IV cannulations which were all reported by one systematic review each. CNS, central nervous system; GI, gastrointestinal; IV, intravenous; LMIC, low/middle-income country; UTI, urinary tract infection.

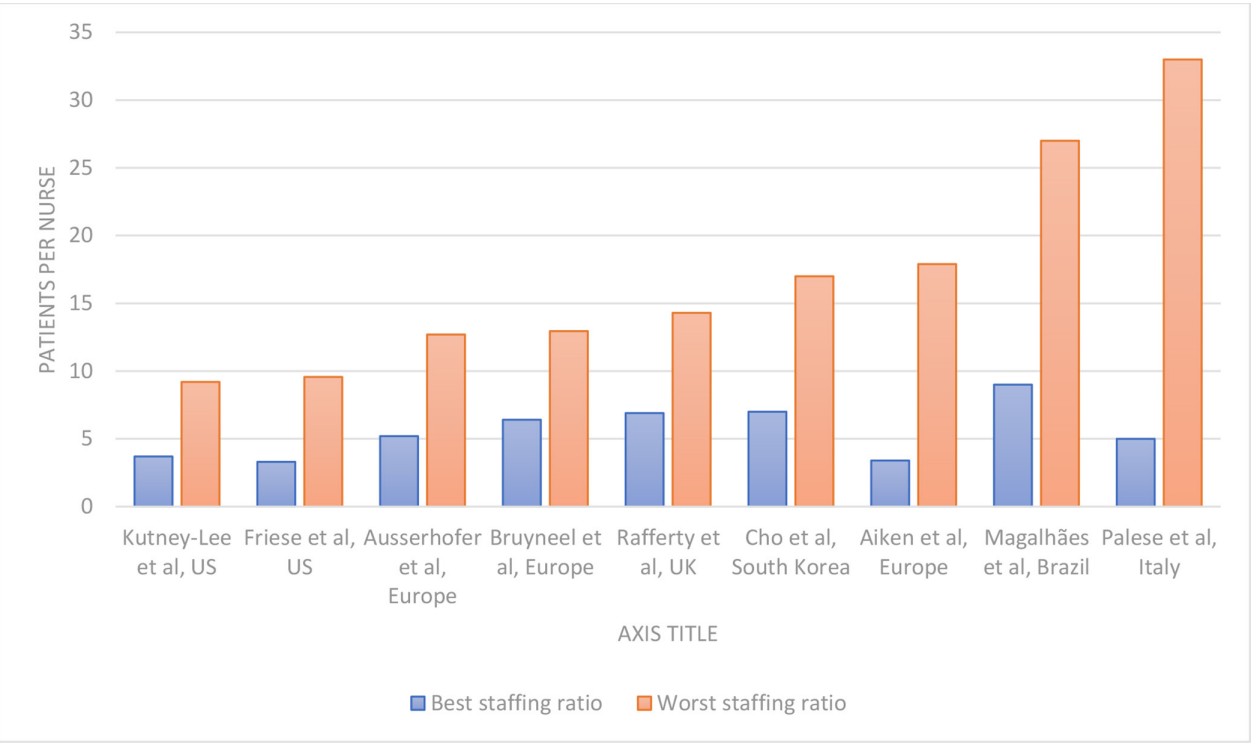

**Figure 4** Bar chart showing the patient-to-nurse ratio per shift range described in individual studies that reported this metric and described a range (countries of origin and first author of the individual studies are presented, and the y-axis represents the number of patients per nurse). The best staffing ratio in the study context is presented in blue while the worst is presented in orange, these best and worst ratios were used in study-specific exploratory of the association between staffing and nursing quality.

staffing can demonstrate positive effects. We noted nurse staffing metrics reported in LMICs were more extreme in comparison with HICs. For example, patient-to-nurse ratio, a metric that measures the number of patients per nurse on a shift, was reported to be as high as 27 patients in a Brazil study,[51] and this starkly contrasts with figures of between four and nine patients per nurse from US studies.[44 49] Also, NHPPD, which measures the total number of nursing hours provided in a defined period adjusted for the number of admissions in the same period, ranged between 4.25 and 5.33 in LMICs,[63 64 71] in contrast with some values from the USA that were 2.5-fold higher.[65] Because the nurse-to-patient ratios are comparatively poorer in LMICs, increasing nurse staffing might potentially lead to larger impacts on quality of patient care and overall patient outcomes. Local research in these contexts is however needed to confirm this.

Staffing levels in the literature were measured using various proxy metrics. While NHPPD and patient per nurse ratios were the most frequently employed staffing metrics across studies, we observed wide heterogeneity in reported staffing metrics across studies. In total, there were 29 reported unique nurse staffing metrics across the literature, and report of their summary statistics was inconsistent across studies. This limited our ability to collate findings across multiple studies to examine the range of nurse staffing where this research has been conducted. Previous reviews have also documented

similar heterogeneity in nurse staffing research and literature.[14] Progress within this area of research would benefit from a move towards standardisation and reporting of core metrics across different studies and research settings to promote comparative learning across contexts.

As a proxy for the quality of patient care, a range of outcomes was reported across systematic reviews. Together, all our included reviews reported 23 different outcomes with the most popular being patient mortality, pressure ulcers and the incidence of hospital-acquired infections. Only eight of these outcomes were reported in studies from LMIC settings. The reported patient care outcomes in LMICs were missed nursing care, mortality, the incidence of pressure ulcers, patient length of stay, treatment errors, hospital-acquired infections and falls. Other outcomes such as incidence of deep venous thrombosis or postoperative complications were not reported. One reason might be limited diagnostic capacity in LMICs; many patient complications may go unrecognised and so the burden of these is under-recognised with no data collected. This finding probably also reflects limited health administrative and secondary data sources in LMICs which preclude the reporting of these data. Indeed, studies within these contexts have been based on reviewing paper-based case records which are usually unstructured and have not been curated for research purposes.[51 76] There are now some examples of networks collecting secondary data in LMICs. For example, in Kenya

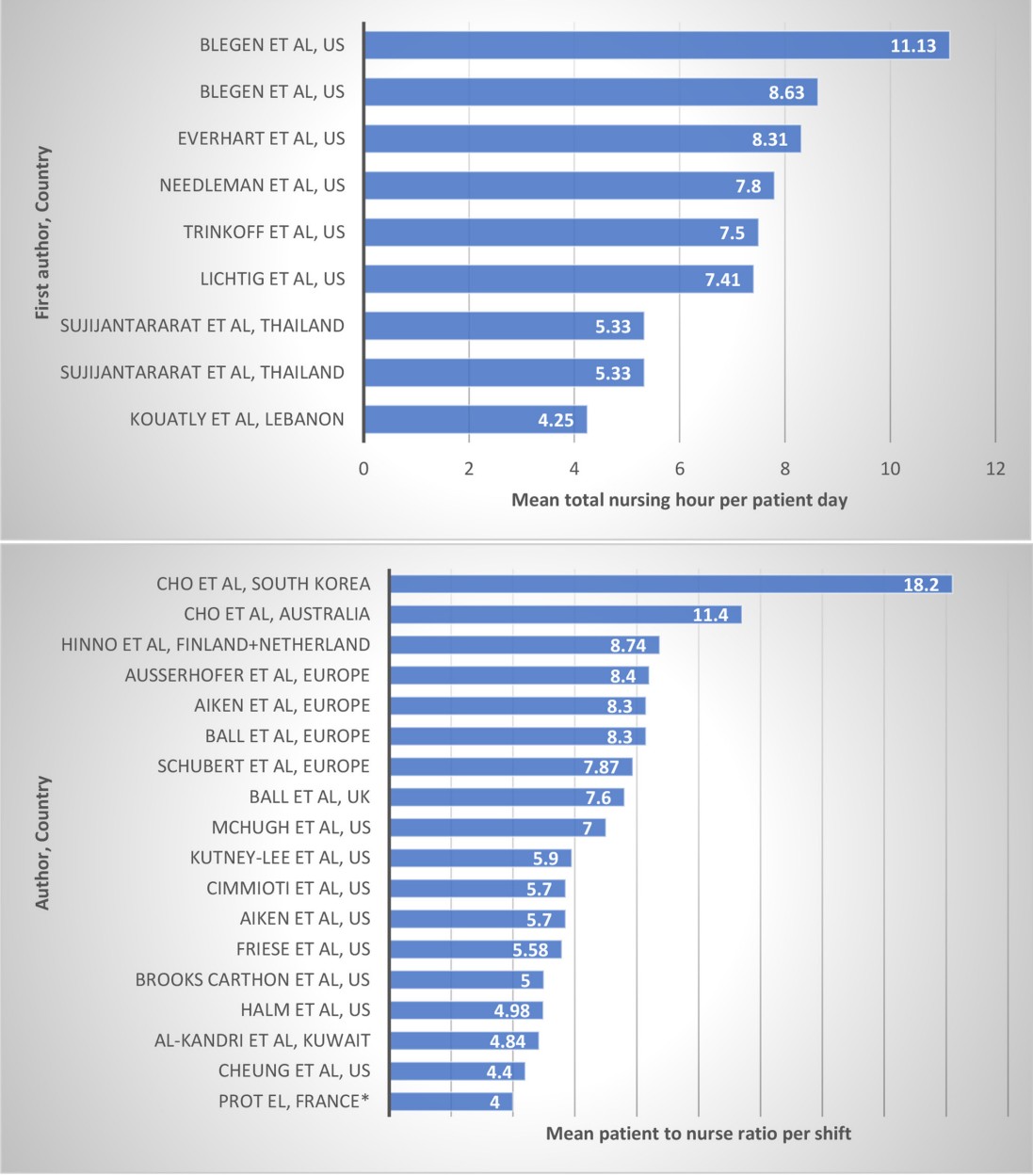

**Figure 5** Horizontal bar charts showing the mean total nursing hour per patient day (top) and the mean patient-to-nurse ratios (bottom) across primary studies reporting these metrics (first author and countries in which individual studies were conducted are depicted on the y-axis). *Median value.

since 2013, the Clinical Information Network collects secondary data using structured admission records from a network of 23 hospitals, and these data have been used repeatedly in quality improvement research,[78] including tracking patient quality indicators such as the frequency at which patient vital signs were conducted.[79] Such databases might play a role in reducing the evidence gap and are likely to be crucial in guiding research to policy. Interestingly, one outcome which does not need advanced diagnostics or secondary data collection is patient satisfaction, but there was no report linking nurse staffing to patient satisfaction from an LMIC.

The majority of the reported outcomes within reviews (22 out of 23) were outcome-based measures of quality of care, except for missed nursing care, a care outcome that describes partially or completely omitted or delayed nursing care[11] which we identified as the sole process-based measure of the quality of care. Care processes occur more frequently than outcomes and as such are useful as studies employing them might be of more manageable size. They might also represent important measures for intervention research aimed at either demonstrating the influence of enhancements in nurse staffing or determining the likely influence of an intervention on the quality of patient care due to their likely proximal position in the causal pathway of quality of patient care.

## STRENGTHS AND LIMITATION

To the best of our knowledge, this is the first review to provide a comprehensive review of the evidence for LMICs in published systematic reviews on the association between nurse staffing and patient care outcomes. We highlight critical gaps in these care settings that need to be the focus of future research in this area.

We recognise that the quality of the information provided by this umbrella review is highly dependent on the underlying quality of the systematic reviews. Some of our included systematic reviews scored low on our risk of bias scores, but because our focus was a broad description of the global evidence to identify gaps for LMICs, we still include these reviews in our synthesis. Due to limitations in translation on the research team, we only considered systematic reviews published in English. We also did not consider qualitative reviews as our umbrella review focused on reviews that investigated an association between nurse staffing and patient care outcomes.

## CONCLUSIONS

Our umbrella review demonstrates that the evidence examining links between nurse staffing and patient care outcomes in LMICs is very limited, both in terms of the number of studies and the outcomes that have been investigated. While there is research conducted in HICs, these might be poor proxies for LMIC settings as staffing ratios in these settings are quite different. Research in LMIC settings is clearly needed to examine the effect of nursing shortages and interventions addressing these on the overall quality of patient care and inform local staffing policy in these regions. To enhance learning across contexts, there needs to be greater uniformity and standardisation of metrics used to measure nurse staffing.

**Author affiliations**
[1]Health Services Unit, KEMRI-Wellcome Trust Research Programme, Nairobi, Kenya
[2]Nuffield Department of Medicine, University of Oxford, Oxford, UK
[3]Centre for Evidence-Based Intervention, Department of Social Policy and Intervention, University of Oxford, Oxford, UK
[4]Department of Paediatrics and Child Health, University of Nairobi, Nairobi, Kenya
[5]Wellcome Trust Research Program, KEMRI-Wellcome Trust Research Programme, Nairobi, Kenya
[6]Centre for Maternal Adolescent Reproductive and Child Health, London School of Hygiene and Tropical Medicine, London, UK

**Contributors** AI conceptualised the idea for the manuscript with inputs from DG, JMM, JA and ME. AI and SO conducted the methods for the paper. AI drafted the manuscript with significant contribution from all authors and under the supervision of DG, JMM, JA and ME. AI is the guarantor of this manuscript and is responsible for the overall content. All the authors reviewed all versions of the manuscript and agreed on a final version for submission.

**Funding** AI is supported for a PhD studentship at the University of Oxford by the National Institute for Health Research (NIHR130812). ME and JA receive salary support from a Wellcome Trust Senior Research Fellowship (# 207522) awarded to ME. JMM and DG receive salary support from the NIHR (NIHR130812): Learning to Harness Innovation in Global Health for Quality Care (HIGH-Q) grant using UK aid from the UK government to support global health research. A Wellcome Trust core award to the KEMRI-Wellcome Trust Research Programme (#092654) enables this research.

**Disclaimer** The funders had no role in the study design, data collection and analysis, or preparation of the manuscript. The views expressed in this publication are those of the authors and not necessarily those of the Wellcome Trust, NIHR or the UK government.

**Competing interests** None declared.

**Patient and public involvement** Patients and/or the public were not involved in the design, or conduct, or reporting, or dissemination plans of this research.

**Patient consent for publication** Not required.

**Ethics approval** This review used secondary data from previously published systematic reviews and so ethical approval or patient consent was not required.

**Provenance and peer review** Not commissioned; externally peer reviewed.

**Data availability statement** Our search strategy is available in the supplemental files and a list of both included and excluded studies have been included in our tables and supplemental files.

**ORCID iDs**
Abdulazeez Imam http://orcid.org/0000-0001-5070-3060
Jalemba Aluvaala http://orcid.org/0000-0002-0851-3711
Jackson Michuki Maina http://orcid.org/0000-0001-6874-8929
Mike English http://orcid.org/0000-0002-7427-0826

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
