## [Reviewer comments · BMJ Open]

ARTICLE DETAILS

TITLE (PROVISIONAL)	IDENTIFYING GAPS IN GLOBAL EVIDENCE FOR NURSE STAFFING AND PATIENT CARE OUTCOMES RESEARCH IN LOW AND MIDDLE-INCOME COUNTRIES: AN UMBRELLA REVIEW
AUTHORS	Imam, Abdulazeez; Obiesie, Sopuruchukwu; Aluvaala, Jalemba; Maina, Jackson; Gathara, David; English, Mike

VERSION 1 – REVIEW

REVIEWER	Schultz, Timothy John Flinders Health and Medical Research Institute
REVIEW RETURNED	27-Jun-2022

GENERAL COMMENTS	Thank-you for the opportunity to review the manuscript "IDENTIFYING GAPS IN GLOBAL EVIDENCE FOR NURSE STAFFING AND PATIENT CARE OUTCOMES RESEARCH IN LOW AND MIDDLE-INCOME COUNTRIES: AN UMBRELLA REVIEW". This presents the research literature in the area of nurse staffing, and offers comparisons between research conducted in low and middle income countries and high income countries. While I can see some merit in the approach taken, I also have a number of criticisms. First, I see umbrella reviews as tools to synthesise large bodies of quantitative research and provide guidance around overall effectiveness (eg of a suite of interventions, or in a wide ranging patient population). What has been conducted here seems to me to be more of a scoping review, in which the aim is to map literature. For example, in this study, the types of outcomes, and how they are measured, is an important result from the study, addressing research question b - What patient care outcomes are reported across reviews and how do reported outcomes differ between HICs and LMIC studies? This seems much more aligned to a scoping review method. It is also a slightly unusual approach to report results such as "ten of 14 reviews had no LMIC publications" (line 35, page 2, Abstract), when at least some of the included studies had specific exclusion criteria around LMIC countries. I looked at the inclusion criteria of only two of the included reviews (Twigg et al 2021 and Lankshear et al 2005). Twigg et al only included studies from high income countries, I suspect this may be the case for other reviews as well, although Lankshear did not specify. Incidentally, Lankshear is misspelt as Lankashear throughout the review. The paper also suffers from a lack of clarity. Nurse staffing is never defined, therefore it is not really possible to appraise how relevant reviews and individual studies are. I note the exclusion of Butler et al 2019, which included nurse staffing models, which they defines as "staffing models, staffing levels, skill mix, grade mix, or qualification mix....Staffing levels include nurse-to-patient ratios, hours of nursing
---

	care, use of full-or part-time staff, or both.” (p. 11). Because the review DID include studies of nurse:patient ratios, it seems that Butler et al should have been included. The lack of explicit inclusion and exclusion criteria and the PICO explication compounds the problem. See comment on p.7 line 38. There is also a lack of specificity in the language employed, such as poor staffing, better ratios etc, but these are all relative – eg poor, or better, compared to what? Page 5 – line 40-42 – it is not really clear why some results from studies that have been included in systematic reviews should be compared “with the global literature”, nor how this review can achieve this aim.
--	--

REVIEWER	Matthews, Anne Dublin City University, School of Nursing, Psychotherapy & Community Health
REVIEW RETURNED	25-Aug-2022

GENERAL COMMENTS	This is an excellent article which adds great value to this field. It rightly highlights all the challenges in the field and the gaps. It fills the intial gap of identifying what has been researched on staffing in hospitals in LMICs. As the review focused on studies in hospitals only (which reflects most work in this area) I was wondering if this should be reflected in the title and abstract- hospital nurse staffing? Just a suggestion, and since most research in this field is only in hospitals, then there is probably no need. This does raise another interesting issue for LMICs, where so much care is provided in health facilities such as rural clinics which are not ‘hospitals’ as such- staffing there can often be even more challenged. This si perhaps something else that needs a greater research focus internationally as so much research findings often highlight challenges with staffing and quality in such settings (for maternity care etc) while not studing staffing directly. Overall the researchers are to be congratulated on this article. V few minor typographical errors: page 3, line 48: we identified data from HICs might not be good proxies for LMICs.. add ‘we identified THAT...’ page 6, line 3: resource challenge LMIC .. change to resource-challenged... Many thanks for the chance to review this work which I look forward to seeing published shortly.
---

VERSION 1 – AUTHOR RESPONSE

Reviewer: 1

Dr. Timothy John Schultz, Flinders Health and Medical Research Institute

Comments to the Author:

Thank-you for the opportunity to review the manuscript “IDENTIFYING GAPS IN GLOBAL EVIDENCE FOR NURSE STAFFING AND PATIENT CARE OUTCOMES RESEARCH IN LOW AND MIDDLE-INCOME COUNTRIES: AN UMBRELLA REVIEW”. This presents the research literature in the area of nurse staffing, and offers comparisons between research conducted in low and middle income countries and high income countries.

Query: While I can see some merit in the approach taken, I also have a number of criticisms. First, I see umbrella reviews as tools to synthesise large bodies of quantitative research and provide guidance around overall effectiveness (eg of a suite of interventions, or in a wide ranging patient population). What has been conducted here seems to me to be more of a scoping review, in which the aim is to map literature. For example, in this study, the types of outcomes, and how they are measured, is an important result from the study, addressing research question b - What patient care outcomes are reported across reviews and how do reported outcomes differ between HICs and LMIC studies? This seems much more aligned to a scoping review method.

Response to query:

We thank the reviewer for this comment, and we agree that one way to answer our research question might have been a scoping review. We are of the opinion that this method would however not have been efficient in answering the proposed research questions as the research area on nurse staffing and patient care outcomes is quite expansive.

While umbrella reviews do have a role in providing guidance for overall effectiveness, they have also been used as a more efficient means to synthesize research in subjects where there is an extensive evidence base. Please see examples of this:

Thompson W, Tonkin-Crine S, Pavitt SH, McEachan RR, Douglas GV, Aggarwal VR, Sandoe JA. Factors associated with antibiotic prescribing for adults with acute conditions: an umbrella review across primary care and a systematic review focusing on primary dental care. Journal of Antimicrobial Chemotherapy. 2019 Aug 1;74(8):2139-52.

Sleddens EF, Kroeze W, Kohl LF, Bolten LM, Velema E, Kaspers PJ, Brug J, Kremers SP. Determinants of dietary behavior among youth: an umbrella review. International Journal of Behavioral Nutrition and Physical Activity. 2015 Dec;12(1):1-22.

Also please see instances where researchers have used similar methodology to describe a range of quality indicators or outcomes:

Ramalho A, Castro P, Goncalves-Pinho M, Teixeira J, Santos JV, Viana J, Lobo M, Santos P, Freitas A. Primary health care quality indicators: an umbrella review. PloS one. 2019 Aug 16;14(8):e0220888.

Blume KS, Dietermann K, Kirchner-Heklau U, Winter V, Fleischer S, Kreidl LM, Meyer G, Schreyögg J. Staffing levels and nursing-sensitive patient outcomes: Umbrella review and qualitative study. Health services research. 2021 Oct;56(5):885-907.

Query

It is also a slightly unusual approach to report results such as “ten of 14 reviews had no LMIC publications” (line 35, page 2, Abstract), when at least some of the included studies had specific exclusion criteria around LMIC countries. I looked at the inclusion criteria of only two of the included reviews (Twigg et al 2021 and Lankshear et al 2005). Twigg et al only included studies from high income countries, I suspect this may be the case for other reviews as well, although Lankshear did not specify. Incidentally, Lankshear is misspelt as Lankashear throughout the review.

Response to query

We thank the review for this comment and have now modified this statement to “only 4 of 14 reviews had data on a LMIC publication (line 33, page 2, Abstract). Additionally, we have also indicated the systematic reviews with a restricted search criteria to high income countries within table 1. We have also corrected the typographical error – Lankshear et al throughout the manuscript.

Query

The paper also suffers from a lack of clarity. Nurse staffing is never defined, therefore it is not really possible to appraise how relevant reviews and individual studies are. I note the exclusion of Butler et al 2019, which included nurse staffing models, which they defines as “staffing models, staffing levels, skill mix, grade mix, or qualification mix....Staffing levels include nurse-to-patient ratios, hours of nursing care, use of full-or part-time staff, or both.” (p. 11). Because the review DID include studies of nurse:patient ratios, it seems that Butler et al should have been included. The lack of explicit inclusion and exclusion criteria and the PICO explication compounds the problem. See comment on p.7 line 38.

Response to query:

We thank the reviewer for this comment, and we have created a PICO section similar to our published protocol. Please see page 8, line 153 to 164.

Butler et al did not identify any eligible studies of nurse staffing interventions such as nurse-staffing levels (please 5. Other hospital nurse staffing intervention, page 16 in Butler M, Schultz TJ, Halligan P, Sheridan A, Kinsman L, Rotter T, Beaumier J, Kelly RG, Drennan J. Hospital nurse-staffing models and patient-and staff-related outcomes. Cochrane Database of Systematic Reviews. 2019(4).)

They however described papers which described the impact of various staffing models on patient and staff care outcomes. This included introduction of a special cadre of nurses – advanced or specialist nurses to existing workforce and nurse assistive personnel. These are out of the scope of the current review

Query :

There is also a lack of specificity in the language employed, such as poor staffing, better ratios etc, but these are all relative – eg poor, or better, compared to what?

Response to query:

We have now used more specific languages in place of these terms.

Query:

Page 5 – line 40-42 – it is not really clear why some results from studies that have been included in systematic reviews should be compared “with the global literature”, nor how this review can achieve this aim.

Response to query:

We thank the reviewer for this comment, and we have now provided a stronger justification for our review. Please see page 5, line 99 to 113.

*****Please also see attached report*****

Reviewer: 2

Dr. Anne Matthews, Dublin City University

Comments to the Author:

This is an excellent article which adds great value to this field. It rightly highlights all the challenges in the field and the gaps. It fills the intial gap of identifying what has been researched on staffing in hospitals in LMICs.

As the review focused on studies in hospitals only (which reflects most work in this area) I was wondering if this should be reflected in the title and abstract- hospital nurse staffing? Just a

suggestion, and since most research in this field is only in hospitals, then there is probably no need. This does raise another interesting issue for LMICs, where so much care is provided in health facilities such as rural clinics which are not 'hospitals' as such- staffing there can often be even more challenged. This si perhaps something else that needs a greater research focus internationally as so much research findings often highlight challenges with staffing and quality in such settings (for maternity care etc) while not studing staffing directly. Overall the researchers are to be congratulated on this article.

V few minor typographical errors:

page 3, line 48: we identified data from HICs might not be good proxies for LMICs.. add 'we identified THAT...'

page 6, line 3: resource challenge LMIC .. change to resource-challenged...

Many thanks for the chance to review this work which I look forward to seeing published shortly.

VERSION 2 – REVIEW

REVIEWER	Matthews, Anne Dublin City University, School of Nursing, Psychotherapy & Community Health
REVIEW RETURNED	26-Sep-2022
GENERAL COMMENTS	All revisions undertaken, as shown in 'marked-up' version.